# Image-Guided Intraoperative Assessment of Surgical Margins in Oral Cavity Squamous Cell Cancer: A Diagnostic Test Accuracy Review

**DOI:** 10.3390/diagnostics13111846

**Published:** 2023-05-25

**Authors:** Giorgia Carnicelli, Luca Disconzi, Michele Cerasuolo, Elena Casiraghi, Guido Costa, Armando De Virgilio, Andrea Alessandro Esposito, Fabio Ferreli, Federica Fici, Antonio Lo Casto, Silvia Marra, Luca Malvezzi, Giuseppe Mercante, Giuseppe Spriano, Guido Torzilli, Marco Francone, Luca Balzarini, Caterina Giannitto

**Affiliations:** 1Department of Diagnostic and Interventional Radiology, IRCCS Humanitas Research Hospital, Via Manzoni 56, 20089 Milan, Italy; giorgia.carnicelli@humanitas.it (G.C.); luca.disconzi@humanitas.it (L.D.); federica.fici@humanitas.it (F.F.); marco.francone@hunimed.eu (M.F.); luca.balzarini@humanitas.it (L.B.); 2Department of Biomedical Sciences, Humanitas University, Pieve Emanuele, 20072 Milan, Italy; armando.de_virgilio@hunimed.eu (A.D.V.); fabio.ferreli@humanitas.it (F.F.); silvia.marra@humanitas.it (S.M.); luca.malvezzi@humanitas.it (L.M.); giuseppe.mercante@humanitas.it (G.M.); giuseppe.spriano@hunimed.eu (G.S.);; 3Otorhinolaryngology Unit, IRCCS Humanitas Research Hospital, Via Manzoni 56, 20089 Milan, Italy; michele.cerasuolo@humanitas.it; 4AnacletoLab, Department of Computer Science “Giovanni degli Antoni”, Università degli Studi di Milano, Via Celoria 18, 20133 Milan, Italy; elena.casiraghi@unimi.it; 5Environmental Genomics and Systems Biology Division, Lawrence Berkeley National Laboratory, 717 Potter Street, Berkeley, CA 94710, USA; 6Division of Hepatobiliary and General Surgery, Department of Surgery, IRCCS Humanitas Research Hospital, Via Manzoni 56, 20089 Milan, Italy; guido.costa@humanitas.it; 7Department of Diagnostic Radiology, ASST Bergamo Ovest, 24047 Treviglio, Italy; 8Department of Biomedicine, Neuroscience and Advanced Diagnostics (BIND), University Hospital of Palermo, 90127 Palermo, Italy; antonio.locasto@unipa.it

**Keywords:** oral cavity cancer, squamous cell carcinoma, tongue cancer, image-guided surgery, surgical margins, magnetic resonance imaging, intraoperative intraoral ultrasound

## Abstract

(1) Background: The assessment of resection margins during surgery of oral cavity squamous cell cancer (OCSCC) dramatically impacts the prognosis of the patient as well as the need for adjuvant treatment in the future. Currently there is an unmet need to improve OCSCC surgical margins which appear to be involved in around 45% cases. Intraoperative imaging techniques, magnetic resonance imaging (MRI) and intraoral ultrasound (ioUS), have emerged as promising tools in guiding surgical resection, although the number of studies available on this subject is still low. The aim of this diagnostic test accuracy (DTA) review is to investigate the accuracy of intraoperative imaging in the assessment of OCSCC margins. (2) Methods: By using the Cochrane-supported platform Review Manager version 5.4, a systematic search was performed on the online databases MEDLINE-EMBASE-CENTRAL using the keywords “oral cavity cancer, squamous cell carcinoma, tongue cancer, surgical margins, magnetic resonance imaging, intraoperative, intra-oral ultrasound”. (3) Results: Ten papers were identified for full-text analysis. The negative predictive value (cutoff < 5 mm) for ioUS ranged from 0.55 to 0.91, that of MRI ranged from 0.5 to 0.91; accuracy analysis performed on four selected studies showed a sensitivity ranging from 0.07 to 0.75 and specificity ranging from 0.81 to 1. Image guidance allowed for a mean improvement in free margin resection of 35%. (4) Conclusions: IoUS shows comparable accuracy to that of ex vivo MRI for the assessment of close and involved surgical margins, and should be preferred as the more affordable and reproducible technique. Both techniques showed higher diagnostic yield if applied to early OCSCC (T1–T2 stages), and when histology is favorable.

## 1. Introduction

A surgical margin is the apparently healthy tissue around a tumor that has been surgically removed. Most commonly, in oral cavity squamous cell carcinoma (OCSCC), a margin wider than or equal to 5 mm is considered “negative”, a margin between 1 and 5 mm “close”, and a margin less than 1 mm “positive” [1,2]. Radicality and negative margin status represent the main goal of oral cancer surgery, which is the cornerstone of treatment and is currently performed for up to T4a lesions [3]. Success of primary resection is a key prognostic factor, and it ultimately determines the need for adjuvant treatment, either radiotherapy or re-resection [4,5]. Failure to achieve clear margins translates into considerable costs, morbidity, and reduced quality of life of OCSCC patients [6,7,8,9]. Involved margins are known to determine a drop in prognosis as well as worse disease outcomes: a metanalysis from Bulbul et al. (2019) [10] demonstrated that re-resection, both intraoperative and late, carries over a twofold increased risk of local relapse compared to first-time clear resection. Similarly, in a cohort study including 753 patients, Hakim et al. [11] showed that involved and close margins, accounting for 50% of patients, carried up to 40% cancer-specific mortality rate along with a progressive decrease in all survival outcomes. Unfortunately, OCSCC is associated with among the highest rates of incomplete surgical resections: close and positive margins are estimated to occur in 25.5–85% cases, with a striking heterogeneity across studies [2,12,13] (mainly related to variability in surgical margin definition and surgical–pathological approaches).

The issue of incomplete surgical excision is particularly relevant to tongue cancer: the tongue is the most frequently involved subsite in oral cancer, and it carries the highest cause-specific mortality, incidence of skip metastases and extranodal extension, and among the highest rates of incomplete surgical excision [14]. The reason for its complexity resides in clinical and anatomical factors. According to standards of practice, the assessment of surgical margins during resection of tongue cancer is performed on manual palpation by the surgeon, a method which has relatively low sensitivity and remains unreliable [3]. Furthermore, the muscular structure of the tongue and the presence of median and paramedian septa running across muscle bundles offer a so called “low-resistance anatomical route” for early cancer spread, often not visible clinically or at superficial imaging [15,16]. Other factors playing a role in the pathogenesis are the rich vascular and lymphatic supply (so called “T-N tract”), and the presence of nerves coursing close to the tongue and within [16]. Histologic type, tumor budding, and noncohesive growth are important predictive factors of OCSCC recurrence and locoregional spread [17,18].

The standard procedure for the assessment of surgical margins involves manual palpation by the surgeon, who aims at resecting at least 10 mm away from the tumor, or the use of frozen section analysis. Despite having excellent specificity (around 95% in most studies), this latter technique is subject to sampling bias, it is rather expensive, and is not available in many hospitals [19,20,21,22]. To complicate things even further, shrinkage often occurs upon surgical excision, the extent varying according to anatomical subsite (overall estimated around 20%) [23]. It is universally accepted that margins <1 mm are a poor prognosticator, but close margins still constitute a matter of controversy [2,24,25], with some authors suggesting that resection should be >7.5 mm [2], and others who would decrease it to 3 mm [26]. To address this issue, new imaging techniques have emerged as potential tools to guide the surgeon during OCSCC resection: mainly intraoral ultrasound (ioUS)—both during resection and ex vivo—and magnetic resonance imaging (MRI), performed ex vivo [21,27]. These provide a “three-dimensional” assessment of the tumor and nearby structures, an added value with regard to frozen sections [24].

The primary aim of this DTA review is to evaluate the accuracy of ioUS and MRI in the intraoperative assessment of surgical margins of OCSCC and to compare it to the standard of care. The secondary aim is to infer the sources of heterogeneity across the studies and the possible reasons behind these.

## 2. Materials and Methods

This systematic review was conducted according to the standards set in the Cochrane handbook for DTA reviews [28]. Two independent reviewers (C.G. and G.C.) selected and analyzed article types among original articles, randomized controlled trials, and cohort studies from 1 January 2016 to 27 February 2023. This time interval was selected to dismiss studies with outdated ultrasound technology.

The main question of the review was formulated according to the “population–intervention–comparator–outcome” (PICO) scheme. The entire process of methodology, data collection and data selection, assessment of bias and applicability, accuracy, and heterogeneity analysis was performed by using the Cochrane-supported platform Review Manager version 5.4 (Review Manager (RevMan) [Computer program]. Version 5.4. The Cochrane Collaboration, 2020). Article selection was performed by searching on online databases CENTRAL, MEDLINE, and EMBASE; search terms and syntax used during data collection are shown in Table 1. Among the words used were oral “cavity squamous cell cancer” OR “tongue cancer” OR “oral cavity cancer”, AND “intraoperative imaging” OR “magnetic resonance imaging” “ultrasound” AND “image guided”, and “margin assessment”, “surgical margin”.

Additional articles were selected by handsearching—namely, by examining the reference lists and selecting papers among the “cited by articles” and the “related articles”.

### 2.1. Criteria for Data Selection

Studies were excluded when

-Outcomes evaluated were not in line with the main question of the review;-The analysis involved anatomical regions other than the oral cavity (e.g., the oropharynx);-The cutoff set for close margins was not 5 mm;-Papers were short communications, oral presentations, letters to editor, posters, or analysis in the setting of a systematic review;-Studies were performed before 1 January 2016;-Frequency of intraoral ultrasound probes was below 15 MHz;-Studies were not performed on humans;-Articles were not in English language.

### 2.2. Quality Assessment and Data Extraction

Papers were analyzed based on their content and study design; the QUADAS-2 tool was used to evaluate population, index test, comparator test, and study flow according to their individual risk of bias and general applicability. The standard and user-defined signaling questions are displayed in Appendix A.

Data extracted from full-text reviewed papers included the number of participants, number of cases with clear (>/=5 mm) margins, close (<5 mm, >1 mm) margins and involved margins (<1 mm), number of re-resections performed, number of patients undergoing adjuvant radiotherapy, false negative (FN), true negative (TN), false positive (FP), and true positive rates (TP), as well as sensitivity and specificity (when available) in both test and control cohorts. Negative predictive value (NPV) for margins <5 mm in each of the studies was calculated by dividing the number of true negatives by the sum of true negatives and false negatives (NPV = TN/(TN + FN)). Ex vivo measurements were taken as reference for data extraction in the ultrasound group, performed as final step of the sonographic assessment, and comparable to ex vivo MRI.

The reporting of the present systematic review is in line with standards set in the Preferred Reporting Items for Systematic Reviews and Meta-Analyses (PRISMA) guidelines [29].

## 3. Results

Results of the search workflow are summarized in Figure 1 (PRISMA flowchart): 6063 citations were found by database searching; after removal of duplicates, 5989 articles were screened based on title and abstract. This resulted in 52 articles for full-text review, which were further reduced in number after excluding short reports, letters to editor, oral presentations, and abstracts; also excluded were studies in which ultrasound was not used intraorally, or fluorescence-based imaging techniques were examined. A final number of 20 papers were identified for accurate text analysis and data extraction: based on the content, we further excluded 10 papers whose outcomes were not in line with the main question of the study, when the evaluation involved also other anatomical districts such as the oropharynx, when the source of bias was significant, and when the study time flow was not adequate (reasons for each article exclusion are specified in Appendix A). At the end of data collection, ten papers were deemed eligible for the purpose of our DTA review [30,31,32,33,34,35,36,37,38,39]. Accuracy analysis (sensitivity, specificity, and receiver operator curve) was performed for four selected studies, in which additional data were available [30,32,35,37].

### 3.1. Study Characteristics

In six studies, the index test was intraoral ultrasound, both intraoperative and ex vivo in all cases; the remaining four studies analyzed the performance of MRI ex vivo in the setting of image-guided OCSCC surgery; study design was prospective with consecutive enrolment in 7/10 studies; in three cases it was retrospective. In four studies, a test control design with two arms were present. Except for the study from Adriaansens et al. (2020) [30], where ultrasound was used in the surgery of oral vestibule cancer, all the articles investigated the accuracy of image-guided surgery in oral tongue cancer. Population sample ranged from a minimum of 10 participants [34,35,37] to a maximum of 165 cases [39], with most studies having fewer than 50 patients. Tumor stage was T1–T2 in one study, T1–T2–T3 in six studies, and T1–T2–T3–T4a in three studies. In all the studies, the cut-off 

Value for margin involvement was <5 mm. A summary of study characteristics is provided in Table 2.

### 3.2. Bias and Applicability

The QUADAS-2 assessment reported overall low bias concerns (Figure 2; Appendix A). Two studies [33,38] were considered at high risk for selection bias (due to the retrospective nature, inclusion/exclusion criteria not stated, limited T1–T2, no masking applied, patients undergoing sentinel node biopsies prior to surgery, or with multiple OCSCC history) and four studies had unclear selection bias reports. Masking (image interpretation and pathology assessment) was applied in four studies overall. Inclusion and exclusion criteria were not clearly stated in three articles [34,37,38]; the other seven studies had clearly stated criteria, consistent across the different authors.

With regard to the index test, risk of bias was high in one study (due to the reader being unexperienced and not blind to histopathology results), and unclear in three studies. No bias concerns were reported in the evaluation of the reference test (histopathology in all cases). Study timing and flow had unclear bias reports in two studies, mainly due to the retrospective nature of the study. Applicability concerns on patients’ selection were reported in one study [34], and regarding the index test in three studies [33,34,37] (mainly for reasons of inclusion criteria not stated, MR scanner type, and reading). We did not exclude studies at high risk of bias from our review in that a meta-analysis was not intended after full-text analysis and because QUADAS-2 assessment is subject to personal judgement, despite being a standardized tool. Furthermore, the sources of bias never appeared to favor the intervention, nor did they influence the main question of the DTA review [40].

### 3.3. Data Extraction

Table 3 depicts the main characteristics and findings by study. Mean tumor thickness (TT) or depth of invasion (DOI) measured ranged from 4 mm to 8 mm; the difference between tumor thickness measured at imaging and histopathological tumor thickness ranged from 0.4 mm to 9 mm.

Ex vivo measurements were shown to be more accurate in all the studies evaluating the performance of ultrasound [30,31,32,33,36,38], and were therefore taken as reference for data extraction. With regard to the diagnostic accuracy, the overall negative predictive value (NPV) (calculation was possible for all studies) of both imaging techniques for margins <5 mm ranged from 0.50 to 0.91; NPV of ex vivo ultrasound ranged from 0.55 to 0.91, and NPV of ex vivo MRI ranged from 0.5 to 0.90. In one study, the assessment of NPV and accuracy measurements was not possible [39], as all data (imaging margins—pathological margins) were expressed as relative measures: in this study, the mean maximum values for the false positive and false negative values between MRI and pathological sections were 3.21 mm and 1.89 mm, respectively.

Sensitivity, specificity, and positive and negative predictive values (PPV, NPV), as well as accuracy (depicted in the receiver operator curve, ROC—Figure 3) were calculated for four selected studies reporting both number of false negatives and positives [30,32,35,37]. For two of the ten studies included [30,35], authors calculated diagnostic accuracy on a per-slice basis instead of performing a per-patient analysis. We extracted data of sensitivity and specificity accordingly, in that analysis on the total number of slices sectioned was deemed more accurate as estimate. In the study from Heidkamp et al. [35], accuracy was calculated both as pooled estimate and for two readers separately (R1,R2); data are reported for each reader in Figure 3. The sensitivity of imaging according to these studies ranged from 0.07 to 0.75; the specificity ranged from 0.81 to 1.00 (Figure 3). Overall, the mean rate of free margin resection (>5 mm) was 55.8% ± 28 cases when imaging guidance was used; except for two studies that showed clear margins in only 8% and 20% cases, the other studies all had a free margin rate above 55%, up to 92%. In all the studies with test control cohort design, the test cohort always showed higher rates of clear margins, with a mean increase in the rate of free margin resection of 35 ± 13%. Considering the significant heterogeneity in the results reported across the studies, meta-analysis was not performed.

## 4. Discussion

This systematic review summarizes the available evidence on the accuracy of ultrasound and MRI in guiding the resection of OCSCC; no previous works addressing this specific topic have been published before, to our knowledge.

Primary OCSCC margin status has a crucial impact on prognosis [11] and ultimately directs patients to adjuvant therapy [4,5,23]. Despite this, the accuracy of surgical excision remains unsatisfactory, with around a 45% rate of incomplete resections [2,12,13]. As shown by Berdugo et al. (2018) [41], the major inaccuracies in surgery are represented by deep margins and perineural invasion, which constitute important predictors of locoregional control [42], and may confound pT staging in up to 30% of pT1 and pT2 cases.

Frozen section analysis is a widely accepted technique for intraoperative margin assessment [21,43]. Despite its high specificity and positive predictive value, the sensitivity for positive margins is highly dependent on sampling (estimated around 50%) [19]; moreover, definitive histopathology is always required, as false negatives can occur [24,44]. FSA is also expensive, with an average change in patient management of only 0.7% cases and a cost–benefit ratio of 1:12 [22]. In order to ensure complete tumor resection, a technique with high sensitivity and NPV is advocated [24]. Imaging tools—in particular, MRI and ioUS—have been validated in the last decade for the estimation of fundamental disease parameters such as DOI, incorporated also in the latest-released version of TNM staging [45,46]. Accurate estimation of depth of invasion is fundamental for planning therapy and surgical approach: several studies have demonstrated a linear relationship between DOI and the risk of cervical nodal metastases, aiding in directing patients to neck dissection [45,47,48].

### 4.1. Accuracy of Ex Vivo Ultrasound

Intraoral ultrasound is now considered a gold standard for the preoperative assessment of OCSCC and for the study of deep margins [49,50,51,52,53]. In a meta-analysis from Nulent et al. [54], ioUS demonstrated high accuracy with an overall overestimation from pathological measurements of only 0.5 mm. Compared to the other techniques, ultrasound offers significant advantages: it is a dynamic examination, noninvasive, and less expensive than frozen section analysis [44]. Measurement of DOI and TT are demonstrated to correlate very accurately with histopathological measures, to the extent that ioUS has been validated [40,52].

In the studies included in our systematic review, the reported accuracy of ex vivo ultrasound was extremely heterogeneous, with negative predictive values ranging from 0.5 [37] to 0.91 [38]. This testifies the scarce reproducibility of this technique in real life. These values may be explained by the variability of the operator across studies, who often was not a radiologist (sometimes either a surgeon or a radiology technician). Other reasons may reside in the probes used during the examination, with frequencies ranging from 15 to 20 MHz; we purposedly excluded studies using US frequencies <15 MHz.

Although the NPV of ex vivo US for margins <5 mm was suboptimal, we observed a consistent improvement when US was used instead of standard resection, with an average increase in the rate of free margins of 35% [30,31,32,35]. On accuracy analysis performed in four selected studies [30,33,35,37], ex vivo US and MRI demonstrated high specificity—ranging from 81 to 100%—but extremely variable sensitivity (from 7% to 75%).

In all the included studies, ultrasound was used both in the intraoperative setting and ex vivo, allowing for a comparison of the two techniques: there was a universal agreement across the studies on ex vivo ultrasound being more accurate than in vivo ultrasound. Another universally reported finding concerned ioUS use for deeply extended tumors: in the majority of the studies, the reported accuracy decreased with increasing depth of invasion [30,33,35,39]. This is in line with the current literature reporting that rate of clearance of deep margins has an inverse relation to DOI [55,56]; similarly, studies demonstrate high diagnostic yield of ioUS when performed in tumors having DOI < 10 mm [57,58].

Another consistent finding across included studies was a higher rate of false negatives when unfavorable histological patterns were present, especially noncohesive growth; the same correlation was found when perineural spread was present [31,39].

Other studies, which were not included in the present systematic review for a matter of year of publication and study methodology, report an improvement of surgical margins upon introduction of ioUS guidance. Songra et al. (2006) [59], who evaluated the performance of ioUS on a cohort of 14 patients with oral cavity cancers (all sites), reported a sensitivity for margins <5 mm of 83% and a specificity of 63%; we did not include this study in our analysis for a matter of methodology (intraoperative examination was performed with a 5–10 MHz probe). Similarly, Baek et al. (2008) [60] assessed the diagnostic performance of ioUS (8–10 MHz) on a prospective cohort of 20 patients with T1–T2 lesions: compared to standard resection, there was no statistical difference in the establishment of safety margins; however, ioUS was significantly more accurate in deep margin estimation and allowed for a significant reduction in the rate of incomplete resections.

Despite the significant advantages of ultrasound, some limitations of this technique need to be acknowledged: above all, the operator dependence and the interoperator variability, the inability to provide a panoramic view on the examined regions, and the lower tissue functional characterization compared to MRI.

### 4.2. Accuracy of Ex Vivo MRI

MRI is the technique of choice for the local staging of OCSCC and is widely used also in the estimation of DOI [61,62], incorporated in the latest version of TNM staging [4].

The diagnostic value of ex vivo MRI was addressed in 4 of the 10 studies in our analysis: in two of the presented studies, the accuracy of MRI was excellent, with values of free margin rate of, respectively, 90% and 92% [34,38]; in the study from Steens et al. (2017) [37], results were discordant, with a rate of free margins of only 20%, despite the use of a high-resolution 7T MR scanner. The authors concluded that the negative predictive value of MRI remains low, despite this technique carrying a good specificity for OCSCC. Weaknesses of this study may reside in the technical inexperience of readers, who had to confront pathologists to improve their performance. In the fourth study by Zhang et al. (2022) [39], individual data from patients were not reported, but MRI was demonstrated to carry a good accuracy, with a mean false positive margin across the 165 cases of 3.21 ± 1.93 mm and a false negative mean margin of 1.89 ± 1.35; in this study, factors associated with higher rates of inaccuracy were, again, DOI and histologic subtype. In the studies from Zhang (2022) and from Heidkamp et al. (2020) [35,39], authors concluded that MRI still carries some limitations, preventing it from being implemented in the clinical practice, and that US currently offers a valuable alternative.

Considering the four studies analyzed in our review, the overall negative predictive value of MRI for margins <5 mm ranged from 0.50 to 0.90, a result which was superimposable to the performance of US. Compared to ultrasound, however, MRI carries some important limitations: first, this technique relies on precession of protons, which can be influenced in a number of pathological conditions. Changes in blood supply, edema, and local inflammation may alter the characteristics of perilesional tissue, producing high rates of false positives (tumor overestimation) [35]. One way to overcome this source of bias may be through the use of diffusion-weighted imaging and apparent diffusion coefficient calculation, which may help distinguishing tumor tissue from perilesional edema [63,64,65].

As second point, characterization of lesions may not be possible due to the limited spatial resolution of MRI: in the study from Steens et al. (2017) [37], margins under 3 millimeters of thickness were not assessable at all on MRI (30% of the total cases). MRI is also expensive and time-consuming (on average 15–20 min) [39], requiring the patient to stay under general anesthesia for a prolonged time. When using MRI to guide OCSCC resection, the surgical and imaging workflow should be carefully planned (MR scanner should be close to the operating room, radiologists should communicate in real time with surgeons, immediate scanning), something which is not feasible in all facilities. The specimen must be reoriented before being scanned, and shrinkage of the specimen upon resection, which is greater at increasing depth of infiltration, often hampers a correct imaging evaluation [35].

Some differences in the technique and methodology should be mentioned: magnet field strength varied from 1.5 to 7 Tesla across the studies. As a second note, masking was not performed in one study, as interpreting radiologists were not blind to histopathology [37]. In addition, no control groups were introduced in any of the studies evaluating the accuracy of MRI.

### 4.3. Potential Sources of Bias and Heterogeneity

We observed considerable heterogeneity in the results of the studies, both in the US cohort and in the MRI cohort. These can be largely explained by the variability of the methods employed: to give one example, the type of surgery varied across the studies and also the number of operators, impacting the rate of success and margin clearance. In many studies, the surgical technique (for instance, wide resection instead of compartmental surgery) was not specified. In the same way, the technique for ioUS assessment and the expertise of the operator varied consistently across studies, as no reference criteria exists yet. Considering these sources of heterogeneity, it appears clear that the intraoperative imaging assessment of OCSCC remains a nonstandardized technique, and establishing a reference method for performing it would provide major improvements in accuracy. Variability in the results of the included studies did not allow for a meaningful meta-analysis. The high number of included articles, however, provided a high-level overview of major themes.

### 4.4. Limitations

This systematic review has limitations, including publication and reporting bias. We did not include studies with unpublished data or preprint studies. We only considered papers concerning US or MRI. Studies that did not have a clear methodology were discarded. As second remark, most of the studies were characterized by a small sample size, and study design was often retrospective for control cohorts compared to test groups.

## 5. Conclusions

The evidence acquired in this systematic review suggests that the NPV of intraoperative imaging guidance for OCSCC margins <5 mm remains suboptimal, with an overall range of 0.5–0.9; this implies around 40–50% involved margins being missed. Despite this, a consistent improvement in free margin resection rate was observed when ultrasound was used (around 35% increase in number of clear margin resections compared to standard surgery). In other words, imaging guidance would always benefit resection, especially for T1–T2 cancers. We found that ex vivo US examination had a comparable accuracy to that of MRI and was preferable in that it is more affordable and reproducible across studies.

Overall, intraoperative ultrasound is still not ready to be fully implemented in the surgical practice but it can provide optimal diagnostic accuracy if some limitations are applied (it is ideal for early OCSCC; sensibility decreases when unfavorable histologic subtypes and noncohesive growth are present).

It must be acknowledged that many published papers have varying methodological quality and limited potential for generalizability and clinical implementation. Many studies are at a moderate risk of bias due to a lack of external validation and systematic guidelines for study design. Herein, we identified some avenues for future research which have the potential to improve the diagnostic yield of ioUS in the evaluation of surgical margins of OCSCC.

## Figures and Tables

**Figure 1 diagnostics-13-01846-f001:**
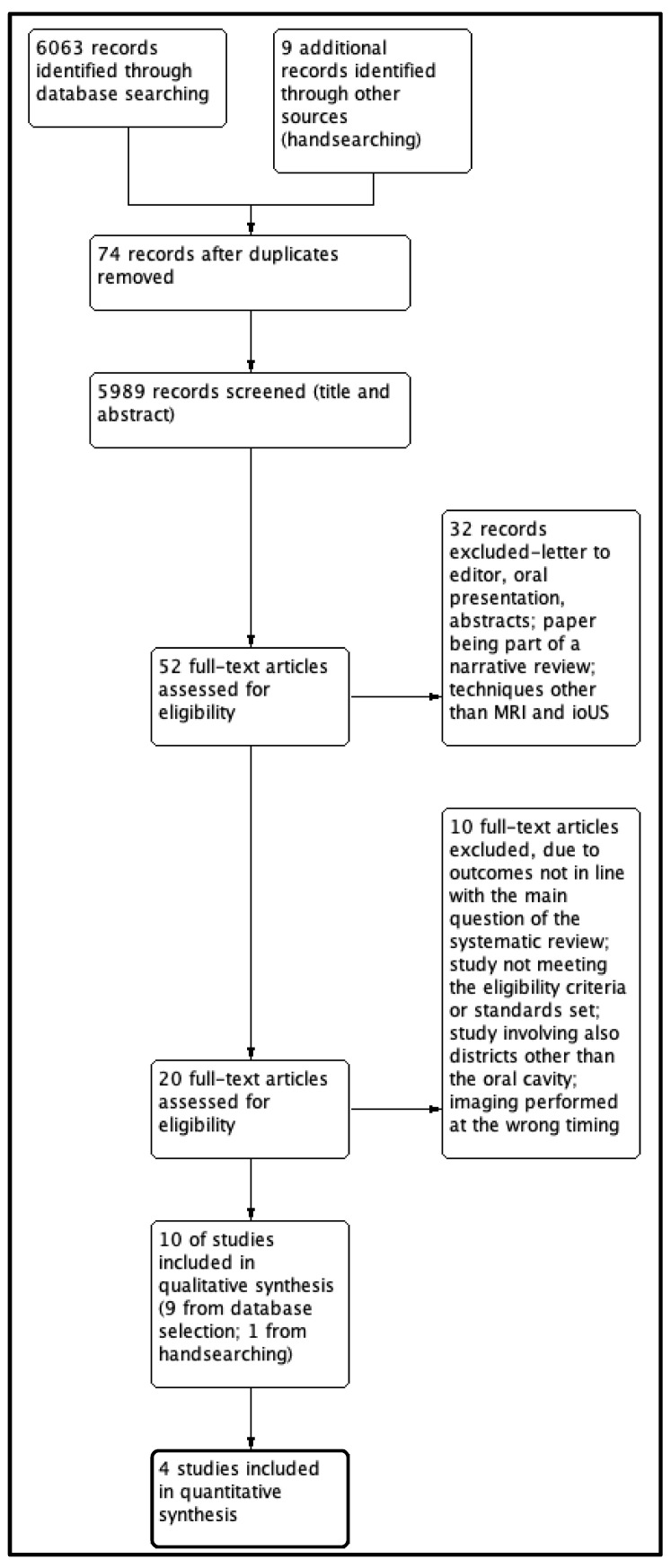
PRISMA. Study flow diagram.

**Figure 2 diagnostics-13-01846-f002:**
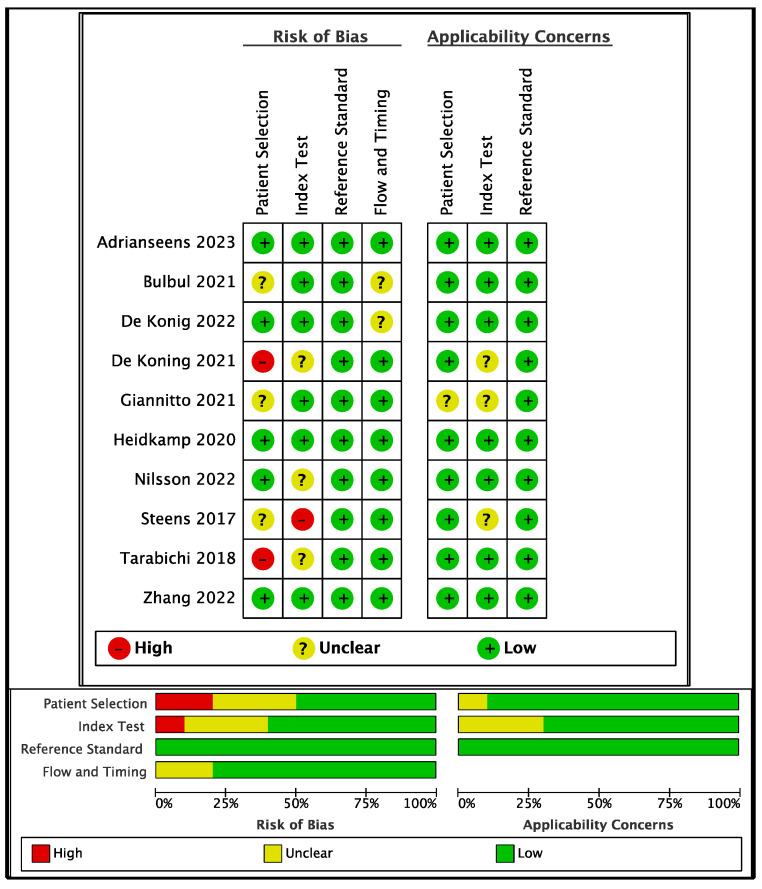
Methodological quality summary and graph (QUADAS-2) [30,31,32,33,34,35,36,37,38,39].

**Figure 3 diagnostics-13-01846-f003:**
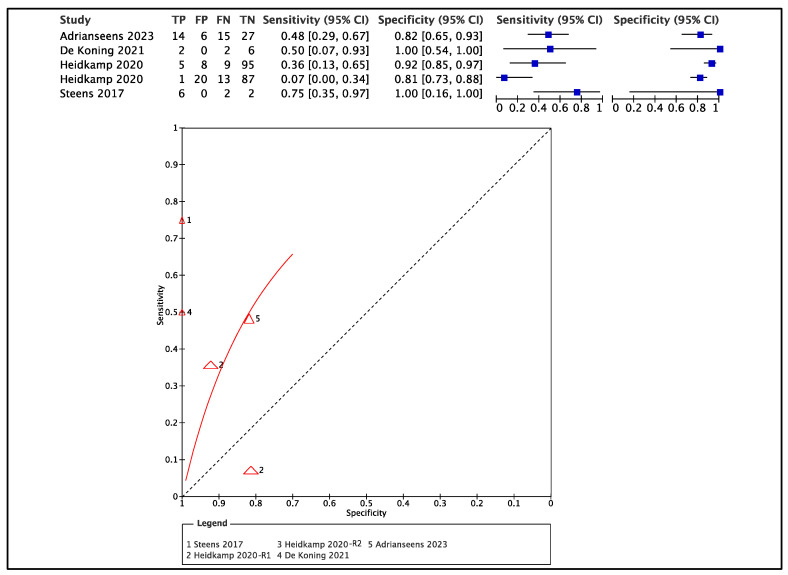
Forest plot of accuracy analysis of four selected studies [30,32,35,37] and relative receiver operator curve. In the study from Heidkamp et al. (2020) [35] data are summarized for Reader 1 (R1) and Reader 2 (R2) separately. Data from Adriaansens [30] and from Heidkamp [35] et al. refer to per-single-slice analysis.

**Table 1 diagnostics-13-01846-t001:** Search terms and syntax used during data collection on online databases CENTRAL, MEDLINE, and EMBASE.

Literature Sources	Search in	Limits	Search Terms
MEDLINE	Advanced Search	Research articles Years (2016–2023) English language Humans	(margin* or specimen or ex-vivo or intraoperative or intra-operative) AND (oral* OR oral cavity or tongue) AND (CANCER OR TUMOUR OR TUMOR OR NEOPL*) AND (MRI Or MR OR magnetic resonance imaging OR nuclear magnetic resonance OR US OR ULTRASOUND OR ULTRASONOGRAPHY) ((‘oral cavity squamous cell carcinoma’ OR ‘oral cavity cancer’ OR ‘tongue cancer’) AND ‘magnetic resonance imaging’ OR ‘ultrasound’) AND ‘margin*’ AND [humans]/lim AND [english]/lim AND ([embase]/lim OR [medline]/lim OR [preprint]/lim OR [pubmed-not-medline]/lim) AND [2016–2023]/py
EMBASE	Advanced Search	Research articles Years (2016–2023) English language	((margin* OR ‘specimen’/exp OR ‘ex vivo’/exp OR intraoperative OR ‘intra operative’) AND (oral* OR ‘mouth cavity’ OR tongue)) AND (cancer OR tumour OR tumor OR neopl*)) AND (mri OR mr OR ‘nuclear magnetic resonance’ OR ‘nuclear magnetic resonance imaging’ OR us OR ultrasound OR ultrasonography)
CENTRAL	Advanced Search	Research articles Years (2016–2023) English Language	((margin* OR ‘specimen’/exp OR ‘ex vivo’/exp OR intraoperative OR ‘intra operative’) AND (oral* OR ‘mouth cavity’ OR tongue)) AND (cancer OR tumour OR tumor OR neopl*)) AND (mri OR mr OR ‘nuclear magnetic resonance’ OR ‘nuclear magnetic resonance imaging’ OR us OR ultrasound OR ultrasonography)

**Table 2 diagnostics-13-01846-t002:** Characteristics of the included studies.

Study ID	Study Design	Anatomical Site	Index Test	Exclusion Criteria	T Stages Included	Control Cohort	Was Masking Applied	Sample Size	Excluded/ Dropouts
Ex-vivo ultrasound									
Adriaansens [30] 2023	Prospective cohort study	Buccal mucosa	ioUS intraoperative and ex-vivo 16 MHz	Preoperative MRI not available (<1 month) tumor out of reach of the IOUS probe	T1-T2-T3-T4a	No	Unclear	14	1
Bulbul [31] 2021	Retrospective case series study with Test and control groups	Oral tongue	ioUS intraoperative and ex-vivo 15 MHz	T4 disease	T1-T2-T3	Yes	Yes	Test group: 23 Control group: 21	0
De Konig [32] 2022	Prospective-nonrandomized trial—consecutive enrolment Test-control cohorts (test prospective; cohort retrospective)	Oral tongue	ioUS intraoperative and ex-vivo 16 and 20 MHz	T4 disease No tumor cells in the specimen Surgery not performed under general anesthesia tumor out of reach of the IOUS probe	T1-T2-T3	Yes	Unclear	Test group: 44 Control group: 96	4
De Konig [33] 2021	Prospective nonrandomized trial—consecutive enrolment Test-control arms (test prospective; cohort retrospective)	Oral tongue	ioUS intraoperative and ex-vivo 16 and 20 MHz	T4 disease Surgery not performed under general anesthesia Previous excisional biopsies or surgery	T1-T2-T3	Yes	Unclear	Test group: 10 Control group: 98	0
Nilsson [36] 2022	Prospective-nonrandomized trial—consecutive enrollment Test-control cohorts (test prospective; cohort retrospective)	Oral tongue	ioUS intraoperative and ex-vivo 18 MHz	Previous surgery/chemo/radiotherapy T4 disease Not suitable for surgery	T1-T2-T3	Yes	Unclear	Test group: 34 Control group: 76	0
Tarabichi [38] 2018	Retrospective cohort study	Oral tongue	ioUS intraoperative and ex-vivo 15 MHz	Not stated	T1-T2	No	No	12	0
Ex-vivo MRI									
Giannitto [34] 2021	Prospective-nonrandomized trial—consecutive enrolment	Oral tongue	ex vivo MRI (1.5T)	Not stated	T1-T2-T3	No	Yes	10	0
Heidkamp [35] 2020	Prospective nonrandomized trial—consecutive enrollment	Oral tongue	Ex-vivo MRI (3T)	Previous surgery/chemo/radiotherapy	T1-T2-T3-T4a	No	Yes	10	0
Steens [37] 2017	Prospective-nonrandomized trial—consecutive enrolment	Oral tongue	Ex-vivo MRI (7T)	Not stated	T1-T2-T4a	No	No	10	3
Zhang [39] 2022	Retrospective cohort study	Oral tongue	Ex-vivo MRI (1.5T)	Previous surgery/chemo/radiotherapy T4 disease	T1-T2-T3	No	Yes	165	0

Abbreviations: ioUS = intraoral ultrasound; MRI = magnetic resonance imaging.

**Table 3 diagnostics-13-01846-t003:** Results of data extraction.

Study ID	Mean TT or DOI	Mean TTΔ Imaging-Pathology	Free Margin Rate	Clear Margins (>5 mm)	Close Margins (1–4.9 mm)	Positive Margins (<1 mm)	NPV	Image-Guided Re-Resection	Reported * Sensitivity, Specificity, AUC	Potential Sources of Bias	Notes
Ex-vivo ultrasound											
Adriaansens [30] 2023	8.6 mm (TT)	1.7 mm in vivo; 1.6 mm ex vivo	8%	1/13 (8%)	9/13 (69%)	3/13 (23%)	0.64 *	Yes (7/13 patients; 20/62 sections)	Sensitivity: 48% Specificity: 82%	Small sample size Short follow-up Surgeon refused re-resection in some cases	Authors suggest surgical margins >7.5 mm The positive margin rate increases with DOI
Bulbul [31] 2021	6.4 mm test vs. 10.8 mm control (DOI)	0.4 mm ex vivo	70% (T) vs. 48%(C)	Test: 16/23 (70%) Control: 10/21 (48%)	Test: 7/23 (30%) Control: 11/21 (52%)	Test: 0/23 (0) Control: 1/21 (4%)	0.69	No	Not reported	Higher number of Node-positive disease in the control group DOI was smaller in the test group Higher number of T2-T3 in the control	The positive margin rate increases with DOI
De Konig [33] 2021	6.2 mm (DOI) vs. 6.1 mm (DOI)	1.9 mm in vivo; 1.4 mm ex vivo	70% (T) vs. 17%(C)	Test 7/10 (70%) Control: 15 (17%)	Test: 2/10 (20%) Control: 67 (74%)	Test: 1/10 (10%) Control: 9 (10%)	0.75	Yes (20%)	Not reported * Extracted: Figure 3	IOUS not performed by a radiologist (technician or surgeon) Inclusion of 8 patients who had undergone a sentinel lymph node procedure with peritumoral injection of a radiotracer (one day prior to surgery) and 1 patient with 2 previous cancers of the oral tongue Disproportionate control group (98 patients)	Authors suggest surgical margins >10 mm The positive margin rate increases with DOI ioUS lowered the rate of close margins; no impact on positive margins
De Konig [32] 2022	7.1 mm (DOI) vs. 7.8 mm (DOI)	0.4 mm in vivo; 0.9 mm ex vivo	55% (T) vs. 16% (C)	Test: 22/40 (55%) Control: 15/96 (16%)	Test: 16 /40 (40%) Control: 67/96 (70%)	Test: 2/40 (5%) Control: 15/96 (16%)	0.55	Yes (33%)	AUC: 0.63	4 dropouts Disproportionate control group (96 patients)	ioUS lowered the rate of adjuvant RT (from 21% control to 10% test group) No statistical difference in the rate of re-resection >3-fold increase in free margin status compared to the conventional cohort (*p* < 0.001)
Nilsson [36] 2022	NA	1.4 mm	55% (T) vs. 28% (C)	Test: 19/34 (55%) Control: 22/76 (28%)	Test: 14/34 (41%) Control: 45/76 (59%)	Test: 1/34 (2%) Control: 9/76 (11%)	0.55	Yes	Not reported	Small sample size	Authors suggest surgical margins >10 mm (accuracy profile too low for closer margins)
Tarabichi [38] 2018	5.45 mm (TT)	1 mm	92%	11/12 (92%)	1/12 (8%)	0	0.91	No	Not reported	10/12 cases were T1, 2 T2 and no T3 No inclusion criteria clearly stated, nor masking Retrospective cohort design	Average margin clearance was 9.8 ± 1.18, with Overtreatment in 41% cases
Ex-vivo MRI											
Giannitto [34] 2021	NA	NA	90%	9/10 (90%)	1/10 (10%)	0	0.90	No	Sensitivity: 90% Specificity: 100%	Small sample size Exclusion criteria not stated	No comparison between histological and pathological margins Average tumor thickness, difference between MR and pathology not stated
Heidkamp [35] 2020	Median 8 mm (DOI)	NA	70%	7/10 (70%)	3/10 (30%)	NA	0.75*	No	Sensitivity R1: 36% Specificity R1: 92% Sensitivity R2: 7% Specificity R2: 87%	Small sample size 50% population having T1 tumors	Poor inter-reader agreement MR-guided resection still not recommended in clinical practice
Steens [37] 2017	4 mm (DOI)	0.45 mm	20%	2/10 (20%)	7/10 (70%)	1/10 (10%)	0.50	No	Not reported * Extracted: Figure 3	Small sample size No masking applied: the radiologist reviewed the images with the pathologist T4 included	DOI < 3 mm could not be assessed Procedural times are too long for clinical application 7T scanner not available in routine practice
Zhang [39] 2022	NA	NA	\	Not applicable	Not applicable	Not applicable	NA	No	Not reported	Retrospective study	consistency of MRI and postoperative pathology cannot be guaranteed. Errors caused by inevitable tumor tissue shrinkage The positive margin rate increases with DOI

* Negative value was calculated on single slices (not on the entire specimen) in these cases. Abbreviations: TT = tumor thickness; DOI = depth of invasion; mean TTΔ = difference between tumor thickness measured on imaging and on pathology; NPV = negative predictive value; AUC = area under the curve of receiver operator curve.

## Data Availability

The data presented in this study are available in the “Appendix A” section. No further new data were created.

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
