# Peer review of "Image-Guided Intraoperative Assessment of Surgical Margins in Oral Cavity Squamous Cell Cancer: A Diagnostic Test Accuracy Review"

_diagnostics, 2023, doi:10.3390/diagnostics13111846_

Round 1

Reviewer 1 Report

This manuscript is about the use of intra-operative in-vivo and ex-vivo imaging techniques to assess surgical margins.

Two different techniques are found: US and MRI. Please discuss these seperately, also in Tables and Figures.

In-vivo and ex-vivo assessments are used. These are different techniques. Please discuss sperately.

Please chck spelling of author names.

Introduction

line 61 ... compared to first time right resection.

line 72 and further. Reasons for incomplete resection. It is because surgeons rely on visual inspection and palpation, which are not reliable enough.

Figure 3. Please check studies and accuracy data. Heidkamp mentioned twice. Then add a and b. IT is 2019 or 2020? Adriaansens did not have 62 (TP+FP+FN+TN) image guided patients. Also other studies not correct. 

Author Response

Dear Reviewer, thank you for your very accurate and constructive revision: we were pleased to carefully examine all the points and concerns raised herein.

Reviewer 2 Report

Thank you for submitting this very valuable paper.

This is first systematic review summarizes the available evidence on the accuracy of ultrasound and MRI in guiding the resection of OCSCC.

All previous reports have been based on small numbers of cases and have not yielded sufficient results, but the results of this review are clinically very meaningful.

I think that it can be posted without a simple correction.

Author Response

Dear reviewer,

Thank you for the positive and constructive feedback given to our manuscript: we are very pleased to revise the content with the suggested adjustments. We carefully edited all the points suggested by reviewer 1, and modified the figures and tables accordingly.

Again, thank you for the time dedicated and for appreciating our work,

Our very best regards

Round 2

Reviewer 1 Report

My comments are adequately and sufficiently answered